# Exploitation of SnO_2_/Polypyrrole Interface for Detection of Ammonia Vapors Using Conductometric and Optical Techniques: A Theoretical and Experimental Analysis

**DOI:** 10.3390/s22197252

**Published:** 2022-09-24

**Authors:** Ajay Pratap Singh Gahlot, Ayushi Paliwal, Avinashi Kapoor

**Affiliations:** 1Deshbandhu College, University of Delhi, New Delhi 110019, India; 2Department of Electronic Sciences, University of Delhi South Campus, New Delhi 110021, India

**Keywords:** gas sensor, surface plasmon resonance, semiconductor metal oxide, conducting polymer, multilayer

## Abstract

This study describes the construction of a lab-built Surface Plasmon Resonance (SPR) system for gas sensing applications employing a highly sensitive and trustworthy optical approach. The nanocomposite thin film of tin oxide (SnO_2_) and Polypyrrole (PPy) were prepared for sensing highly toxic gas, i.e., ammonia (NH_3_) gas. The gas sensor was validated by both optical and conductometric techniques of gas sensing. The optical SPR gas sensor is based on the change in refractive index at the SnO_2_/Polypyrrole (PPy) interface with gas adsorption (NH_3_). The thickness of SnO_2_ and Polypyrrole thin films was optimised using theoretical calculations for a sharp SPR reflectance curve. The manuscript also offers theoretical SPR curves for different PPy and SnO_2_ layer thicknesses. To support the theoretical conclusions, the effects of NH_3_ gas on the prism/Au/SnO_2_/Polypyrrole system were also investigated experimentally. In comparison to other research described in the literature, it was observed that the constructed sensor’s sensitivity was higher. The obtained results demonstrate the utility of the SPR setup in the investigation of the interactions of adhered gas molecules with dielectrics and gas sensing. For conductometric gas sensing studies, the film having optimised thicknesses for sharp SPR reflectance curves was separately prepared on Interdigitated Electrodes. At a low working temperature of roughly 150 °C, the sensing response of the constructed film was observed and found to be maximal (60).

## 1. Introduction

The scientific community has been working to develop effective gas sensors for several decades in order to safeguard the environment and workplaces from harmful and deadly gas leaks [1,2]. Technologies for sensing gases come in a variety of forms, such as conductometric and optical sensing. On the other hand, conductometric sensing technology has a number of limitations, such as the requirement for elevated temperature, high energy consumption, and poor selectivity. Therefore, the development of an efficacious gas sensor based on optical sensor systems is the answer to this problem [3]. It is preferable to use optical sensors since they may operate at room temperature and use less energy [4]. Numerous optical techniques, such as infrared spectroscopy [5], ultraviolet-visible spectroscopy [6], ellipsometry [7], and SPR [8], are accessible for sensing applications. Compared to all other methods, optical gas sensors based on SPR have many advantages. SPR-based optical gas sensors provide great benefits over all other approaches, including easy fabrication of samples, sensing operation effective at room temperature, achieves lowest detection limits, good selectivity, and high responsiveness. Using the Kretschmann arrangement, investigators have deposited a gas-sensitive film on top of a metal layer to use the SPR technology for applications involving gas sensing [9,10,11]. However, the response obtained by the developed sensor is poor. As a result, optimizing the thickness of the detecting layer is the first step in developing a gas sensor. The current research begins with a theoretical simulation to determine the optimal thickness of SnO_2_ and Polypyrrole thin films independently. The metal-dielectric interface can undergo even the smallest modifications, and SPR is a sensitive method for detecting these changes [12]. As a result, a particular gas’s presence in the system may be associated with the shift in SPR reflectance curves.

The detection of extremely dangerous and harmful ammonia (NH_3_) gas is the foremost concern of the study presented here. Animal manure or pollutants from manufacturing and vehicles both produce NH_3_ gas. Many efforts are still being made by the research community to detect NH_3_ gas [13,14]. To detect NH_3_ leakage at exceptionally low quantities (less than 5 ppm), investigators are currently developing a sensor system [15]. Only a few studies explain SPR sensing devices that are intended only to identify NH_3_ gas [16,17]. However, the research showed low sensitivity to detect higher ammonia concentrations (>25 ppm). Maciak et al. detected ammonia vapours utilizing a polyaniline sensing layer; 32 ppm of ammonia vapours were found using an SPR spectroscopy method [16]. A separate sensor layer, Nafion/WO_3_, was utilised by Dolbec et al. [17] to detect higher ammonia concentrations. Gas sensing utilizing the SPR approach, as previously described, necessitates an optimum sensing layer for detection. As a result, selecting a good material and optimizing it is critical for the sensor’s development. The concentration of gas molecules interacting with the sensing layer determines how the index of refraction of the layer changes after exposure to the target gas.

Tin oxide (SnO_2_) is the most suited metal oxide for use in gas sensing applications, and it has been shown over the past few decades that semiconductor metal oxides are excellent materials for sensing applications, such as the detection of a variety of hazardous gases. Tin-oxide (SnO_2_) is a semiconductor metal oxide that is commonly employed as a sensor for hazardous gases due to its high sensitivity and stability [18]. The n-type semiconductor material SnO_2_ is thought to be the best contender for developing gas sensors for a number of dangerous gases [19].

Compared to the single component, there are advanced nanostructures with organic/inorganic composites for gas sensing applications [20]. Increased conductivity, porosity, and catalytic activity are the benefits of using composites in applications requiring gas sensing [20]. There are various organic materials that have recently been exploited for gas sensing applications [21,22]. Metal oxide gas sensors initially demonstrate strong sensitivity to harmful chemicals, but they require higher working temperatures. Selectivity and other sensing properties can be improved by using nanocomposite thin films of polymeric materials and metal oxides [23]. Another type of material that has been studied recently is conducting polymers, which have strong electrical and optical properties [24]. Because it is stable at high temperatures and it is simple to prepare samples, polypyrrole (PPy) is the most well-known conducting polymer [25]. PPy thin films for NH_3_ sensing were prepared electrochemically and had concentrations of between 0.01 and 1% [26]. With the use of chemically prepared PPy films, Yadav et al. observed a 4–8% sensing response to NH_3_ gas [27].

The majority of the findings show that at higher temperatures, the sensing response is minimal. There is no published study on how to build a SnO_2_/PPy multilayer-based SPR sensor for NH_3_ gas detection. This could enable the development of a low-level ammonia gas detection SPR sensor that works well.

In this study, the thicknesses of the thin films of Au, SnO_2_, and PPy were theoretically tuned for a sharp SPR curve. The SPR approach was used to examine the change in optical properties of a gas sensitive multilayer thin film structure constructed at the optimum thickness under varied concentrations of hazardous gases, such as NH_3_ gas.

Studies have been carried out on conductometric gas sensing of the optimised PPy/SnO_2_ multilayer thin film structure. The aim is to observe the sensing response using the resistive method for the best optimized film obtained using SPR.


**Theoretical simulation details**


When light is reflected from metal/dielectric surfaces, Fresnel’s equations can be utilised to characterise the behaviour of the reflected light intensity as a function of angle of incidence (angular interrogation mode). Theoretical reflectance values for one-layer (prism + metal + air) and two-layer (prism + metal + dielectric + air) systems, respectively, are used to calculate the dielectric constant (or refractive index) of the metal and dielectric layer [28]. The following is an explanation of the theoretical Fresnel’s equations used to simulate the graphs. For the (prism + Au + air) system, which is a single layer system with a metal (Au) sheet of thickness *d*_1_, the reflectance at an incident angle is given by
(1)$$\;R_{014}={\left|\frac{r_{01}r_{14}\;\exp\left(2ik_{z1}d_{1}\right)}{1+r_{01}r_{14}\;\exp\left(2ik_{z1}d_{1}\right)\;}\right|}^{2}\;$$
(2)$$\mathrm{where}\;r_{ik}=\frac{\left(\frac{k_{zi}}{\varepsilon_{i}}-\frac{k_{zk}}{\varepsilon_{k}}\right)}{\left(\frac{k_{zi}}{\varepsilon_{i}}+\frac{k_{zk}}{\varepsilon_{k}}\right)}\;$$
(3)$$\mathrm{and}\;k_{zi}=\frac{2\pi }{\lambda }\left(\varepsilon_{i}-\varepsilon_{0}\sin^{2}\;\theta \right)\;$$
where *λ* is the wavelength of the incoming light and *ε_i_* with *i* = 0, 1, and 4 are the corresponding dielectric constants of the glass prism, Au film, and air.

Given known values of *ε*_0_ = (1.517)^2^ and *ε*_4_ = 1.0 Equations (1)–(3) were used to obtain the theoretical SPR data, with the imaginary component (11.9 + 1.2 *i*) representing the dielectric loss of the Au thin film and *ε*_1_ (= *ε*_1_’ + *j**ε*_1_”) serving as the dielectric constant. For the (prism + Au + SnO_2_ + air) system, a two-layer system with an Au film of thickness *d*_1_ and a SnO_2_ film of thickness *d*_2_ might provide the predicted SPR graph. For the (prism + Au + SnO_2_ + air) system, the reflectance R_0124_ is given by
(4)$$R_{0124}={\left|\frac{r_{01}+r_{124\;}exp\left(2ik_{z1}d_{1}\right)}{1+r_{01}r_{124\;}exp\left(2ik_{z1}d_{1}\right)\;}\right|}^{2}\;$$
(5)$$where\;r_{124}=\frac{r_{12}+r_{24}exp\left(2ik_{z2}d_{2}\right)\;}{1+r_{12}r_{24}exp\left(2ik_{z2}d_{2}\right)}\;$$
where *r_ik_* and *k_zi_* are determined by (2) and (3), respectively, and ε_2_ and d_2_ are the complex dielectric constant and thickness of the thin SnO_2_ layer, respectively. The subscripts 0, 1, 2, and 4 stand for various media, including, successively, a glass prism, an Au film, a SnO_2_ film, and air. The literature is used to determine the SnO_2_ thin film’s dielectric constant, which is *ε*_2_ (=*ε*_2_’ + *jε*_2_”).

The reflectance for (prism + Au + SnO_2_ + PPy + air) structures, having a three-layer system (Polypyrrole, SnO_2_ thin films and Au layer), is given as
(6)$$\;R_{01234}={\left|\frac{r_{01}+r_{1234}\;\exp\left(2ik_{z1}d_{1}\right)}{1+r_{01}r_{1234}\;\exp\left(2ik_{z1}d_{1}\right)\;}\right|}^{2}\;$$
(7)$$where\;r_{1234}=\frac{r_{12}+r_{234}\;\exp\left(2ik_{z2}d_{2}\right)\;}{1+r_{12}r_{234}\;\exp\left(2ik_{z2}d_{2}\right)}\;$$
where *r_ijk_*, *r_ik_*, and *k_zi_* are supplied by (5), (2), and (3), respectively, while subscript 0, 1, 2, 3, and 4 indicate the various media, including the prism, Au layer, SnO_2_ film, PPy film, and air, respectively. On the surface of the SnO_2_ thin film, *d*_3_ and *ε*_3_ represent the average thickness of the PPy thin film and the complex dielectric constant, respectively. The value of the PPy thin film’s complex dielectric constant (*ε*_3_ = *ε*_3_’ + *jε*_3_”) was obtained from published sources.

## 2. Materials and Methods

This work prepared the film’ multilayer structure. First, using thermal evaporation and the specifications described in our earlier research [29], an optimal Au thin film was formed on a glass prism. The SnO_2_ film was then applied to the Au coated prism. SnO_2_ nanoparticles were synthesised using the sol–gel process, and, subsequently, thin films were created using the spin coating approach [30]. Finally, thin PPy layers were applied to this framework. PPy nanoparticles were synthesized chemically, and films were prepared via a spin coater [30].

Excitation of Surface Plasmon (SP) modes were achieved using a right-angled BK-7 glass prism (n_p_ = 1.517) in Kretschmann configuration. A p-polarized He-Ne laser (=633 nm for the (prism + Au + SnO_2_ + PPy + air system) was employed as the excitation source. To estimate the SPR response, the reflected light was observed as a function of incidence angle. A separate attachment of a gas cell was prepared such that gas vapours directly hit the gas sensitive-coating (prism + Au + SnO_2_ + PPy). The cell was first vacuumed, then a target gas of NH_3_ at a particular concentration was introduced via the intake valve. Both static and dynamic testing procedures were used. The target gas was positioned within the gas cell for the full range of incidence angles, while the SPR reflectance curve was recorded in the static mode using an optical power metre and rotating stage. A CCD camera was utilised for dynamic measurements to record the response’s fluctuation over time (PC interfaced).

The multilayer thin film structure of PPy/ZnO was formed on Platinum (Pt) Interdigitated Electrodes. Using this measurement technique, the variation in the resistance of the sensor was measured in response to the target gas of fixed concentration.

## 3. Results and Discussion

### 3.1. Optimization of SnO_2_ and PPy Thicknesses

Theoretical SPR reflectance curves obtained from Fresnel’s equations as described in the theoretical section for the prepared systems are presented in Figure 1. As the number of layers increased, i.e., when SnO_2_ was deposited on the Au-coated prism and when PPy was deposited on the (SnO_2_ + Au + prism) surface, the resonance angle increased. This was attributed to the variation in the refractive index of the sensing layer (SnO_2_ + PPy layer).

#### 3.1.1. Varying Thickness of SnO_2_

Figure 2 illustrates the SPR reflectance curve for the multilayered (prism + Au + SnO_2_ + PPy + air) system by changing the thickness of SnO_2_. Figure 2 shows that the lowest reflectance and observed fluctuation of resonance angle are both displayed. As noted from Figure 2, the resonance angle increases from 42.8° to 47.5°, whereas the minimum reflectance decreases from 0.08 to 0.01, with an increase in thickness of SnO_2_.

Table 1 lists the values of FWHM with the thickness of the SnO_2_ film. FWHM increases with thickness, having minimum value for the lowest value of thickness. As a result, 100 nm thick SnO_2_ SPR has the lowest FWHM (or absorption losses), making it the ideal SnO_2_ layer thickness.

#### 3.1.2. Varying Thickness of Polypyrrole Thin Film

A similar variation by varying the thickness of PPy is shown in Figure 3a. All the resonance parameters with corresponding variation with thickness of PPy layer are plotted in Figure 3a.

Table 2 shows the values of FWHM with thickness for the PPy layer. As observed from the studies, the system having a 300 nm thick PPy layer with minimum FWHM (or absorption losses) is the optimized system.

### 3.2. SPR Gas Sensing Experimental Studies

Before performing the experimental gas sensing studies on the prepared sample, the characterisation studies, i.e., XRD and AFM, were also included. Figure 4a shows the XRD spectra indicating the crystalline growth of SnO_2_ thin film and PPy thin film having amorphous structure but does not show any peak in the spectra. Using Scherrer’s formula along the prominent (101) peak, the crystallite size of the nanocomposite SnO_2_/PPy thin film with 100 nm and 300 nm thicknesses of SnO_2_ and PPy, respectively, was calculated. It was found to be around 5.2 nm. The surface morphology of the produced sample with uniformly dispersed crystallites is seen in the AFM picture in Figure 4a.The porosity of the sample was also found to be high. Characterisation studies confirmed that the sample was optimized for a higher sensing response due to high porosity and uniformly distributed crystallites. Figure 4a shows the schematic of SPR gas sensor having metal/dielectric interface as (Au + SnO_2_ + PPy), the Multi layered sensing element. Surface Plasmon Wave was generated at the interface of Au/SnO_2_ propagating along the interface. Fixed concentration of NH_3_ gas was exposed to the surface of the (prism + Au + SnO_2_ + PPy) system. Utilizing the optimized system (prism + Au + SnO_2_ + PPy + NH_3_), the gas sensing results were obtained by exposing the sensing with different concentrations of NH_3_ gas (0 to 200 ppm) and the recorded variation is shown in Figure 4b. With a rise in NH_3_ gas concentration from 1 to 200 ppm, a consistent change in the SPR Resonance angle toward a higher angle was seen from 40.7° to 51.1° (Figure 4b). This ongoing change was caused by the changes in the SnO_2_ thin film’s refractive index due to the adsorption of reducing NH_3_ gas at various concentrations.

The resonance parameters (θ_SPR_, R_min_ and FWHM), as evaluated from SPR curves in Figure 4b, are plotted in the respective graphs of Figure 5a–c. Theoretical evidence indicates that the shift in resonance that was seen experimentally is associated with the change in the effective refractive index of the (SnO_2_ + PPy) sensing interface brought on by the adsorption of reductive NH_3_ gas at various concentrations. Up to 10 ppm of NH_3_ gas concentration, both θ_SPR_ and R_min_, linearly increased. When the concentration of NH_3_ gas increased further, they saturated, suggesting that there was no discernible change in the effective refractive index. A similar variation in FWHM of SPR was recorded.

### 3.3. Sensing Mechanism

There are two major factors behind the detection of NH_3_ gas molecules using the (Au + SnO_2_) sensing layer. First, the change in the semi-infinite dielectric medium from air to gaseous molecules with a greater index of refraction for Ammonia (NH_3_). Secondly, an increase in the refractive index reactive layer of gas with a rise in the concentration of adsorbed NH_3_ gas molecules. The basic reason for the rise in the sensing layer’s refractive index, i.e., SnO_2_ is that the SnO_2_ reduced after interacting with NH_3_ gas molecules. After reduction, oxygen molecules were released from the film surface, hence the trapped electrons. This finally led to an increase in the thin-film SnO_2_’s refractive index.

The calibration curve of the prepared sensor, i.e., variation in effective index of (SnO_2_ + PPy) interface versus concentration, is shown in Figure 6a. As noted from Figure 6a, the refractive index linearly increases until 10 ppm. As a result, a different graph is constructed in Figure 6b for the variable refractive index with increasing concentration up to 10 ppm (b). The curve was fitted linearly, and the sensitivity value was determined by the calibration curve’s slope, i.e., 4.5 × 10^−3^ RIU/ppm.

Furthermore, humidity is a crucial factor in assessing the performance of a gas sensor. Hence, to check the effect of humidity on the performance of a fabricated SPR sensor, the gas sensing studies were carried out for a fixed concentration of 10 ppm of NH_3_ gas for varying humidity ranges (40% to 80%). SPR reflectance curves for the prism/Au/SnO_2_/PPy configuration on exposure to 10 ppm concentration of NH_3_ gas under different relative humidity conditions (40 RH% to 80 RH%) are shown in Figure 7a. The curves are found to slightly shift towards a higher angle with a slight increase in minimum reflectance. This is due to a change in the optical property (increase in conductivity) of the SnO_2_/PPy nanocomposite thin film as the humidity level increased. The calibration curves are also shown showing the corresponding variation of θspr and Rmin with the different relative humidity ambient (40% to 80%) in Figure 7b. The sensitivity of the prepared LRSPR sensor towards humidity derived from the calibration curve is 0.002°/RH%, which is negligible.

The dynamic response for the multi-layered system composed of a prism, Au, SnO_2_, and PPy when exposed to NH_3_ gas at concentrations ranging from 0 to 200 ppm is also included in the current work (resonance angle set at 40.7 degrees). A continuous step-like behaviour was observed from 1 to 200 ppm of NH_3_ gas at room temperature (Figure 8a). On the basis of the calibration curve, the constructed SPR sensor has a sensitivity of 0.202 ppm for NH_3_ concentrations ranging from 1 ppm to 200 ppm, (Figure 8b). The results indicate how to design a reliable SPR gas sensor for quick NH_3_ gas detection.

### 3.4. Conductometric Gas Sensing Studies

Response variation of the optimized sensor with annealing temperature is shown in Figure 9 and it is evident from the sensing graph that there is a maximum response of 60 at a temperature of 150 °C. The sensing response is calculated as:S = (R_a_ − R_g_)/R_g_
where R_g_ is associated with resistance in the presence of the target gas and R_a_ is defined as sensor resistance in the presence of clean air. The sensor’s response time is the amount of time it takes to reach 90% of saturated resistance in the presence of reducing (NH_3_) gas, whereas the sensor’s recovery time is the amount of time it takes to reach 10% of saturated resistance. The sensing response shows maxima due to highly porous film having higher surface roughness, leading to maximum adsorption of target gas molecules. The adsorption leads to a change in the resistance of the film showing maxima. The sensor was also tested for repeatable characteristics. Another especially important parameter of the sensor is the selectivity. Selectivity of the sensor is also shown in Figure 9. The developed sensor was found to be highly selective towards the target gas, which is clear from the figure where a low response is observed for other interfering gases and a high response for the target gas. The reason behind the selective sensing of ammonia gas by the prepared sensor is the NH_3_ gas adsorbed on the SnO_2_/PPy surface. The sensor’s ability to be selective in terms of sensing is greatly enhanced by the PPy thin layer and the NH_3_ gas due to a de-doping reaction between PPy and NH_3_ molecules. Hence, Figure 9 shows the highest sensing response of the prepared sensor towards NH_3_ gas.

A lot of work has been conducted on NH_3_ gas sensors using various metal oxide thin films, including ZnO [31,32], etc. Among these metal oxides, SnO_2_ thin film was studied extensively in comparison to other semiconducting materials towards the detection of a number of both oxidizing and reducing gases [33]. SnO_2_ is the most used material for conductometric gas sensors [33]. Very few reports are available on NH_3_ using SPR [16]. However, the reports lack in various context. The sensitivity was reported to be poor and mostly SPR sensors are utilized to identify ammonia with a greater concentration (>25 ppm). There are no published reports of SPR gas sensors for NH_3_ that use SnO_2_ sensing layers, even though SnO_2_ exhibits enhanced sensitivity for most of the gases [16]. This might assist in the development of effective SPR sensors for detecting ammonia gas at low levels. Additionally, an effort has been made to make the sensor more selective by incorporating the PPy layer. The sensor in the present study has higher sensitivity compared to other reports in the literature [34,35].

## 4. Conclusions

An optical gas sensor that is room-temperature operated has been designed for the detection of highly toxic gas, i.e., NH_3_ gas. SPR technology has been successfully exploited using the sensitive sensing interface of SnO_2_ and PPy. Thicknesses of SnO_2_ and PPy were optimized theoretically. The SPR sensor exhibited a quick response time (1 s) and high sensitivity (4.5 × 10^−3^ RIU/ppm) towards NH_3_ gas across a wide variety of concentrations (1 to 10 ppm). Additionally, dynamic behaviour was seen, and the calibration curve’s estimated sensitivity of 0.202/ppm was used. The sensor was also tested for conductometric sensing.

## Figures and Tables

**Figure 1 sensors-22-07252-f001:**
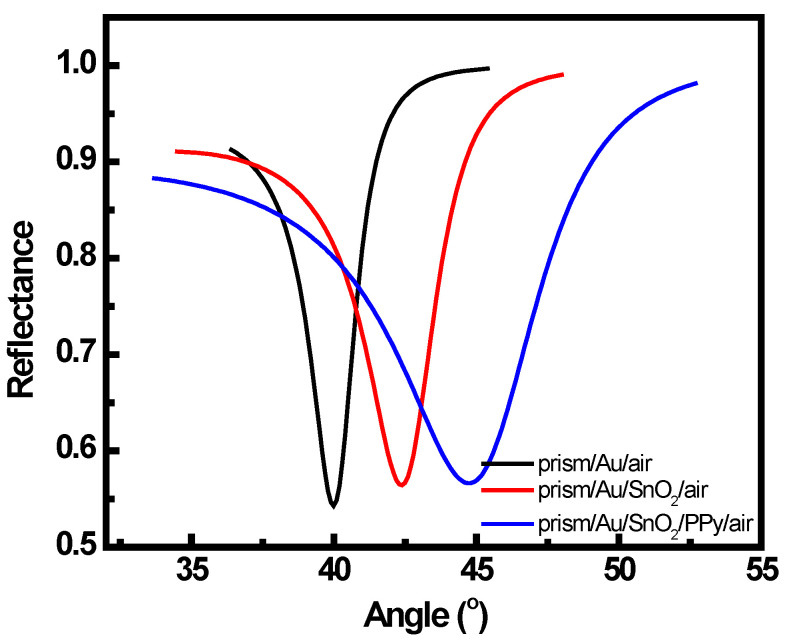
Theoretically simulated SPR reflectance curves for prism/Au/air, prism/Au/SnO_2_/air, and prism/Au/SnO_2_/PPy/air.

**Figure 2 sensors-22-07252-f002:**
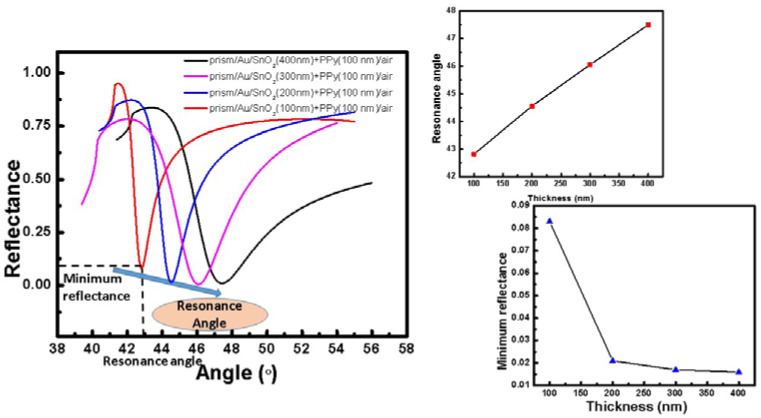
(Prism + Au + SnO_2_ + PPy + air) Multilayered systems’ Theoretically simulated SPR reflectance curves by varying the thickness of SnO_2_ and variation of resonance angle and minimum reflectance with the thickness of SnO_2_.

**Figure 3 sensors-22-07252-f003:**
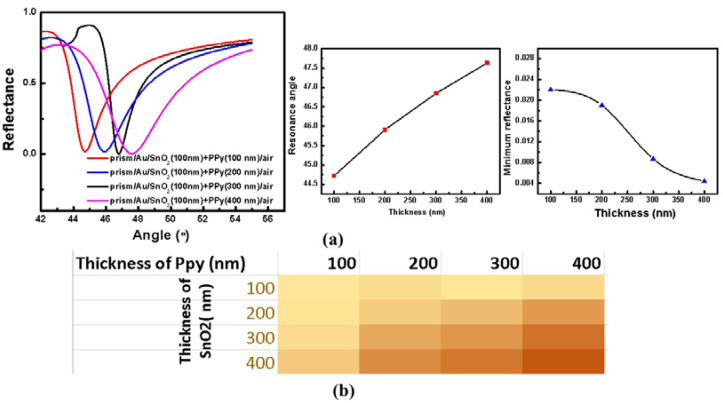
(**a**) (Prism + Au + SnO_2_ + PPy + air) Multilayered systems’ SPR reflectance curves with variable PPy thickness, resonance angle and minimum reflectance with variable SnO_2_ thickness, (**b**) heat map showing the colored variation for optimizing the thicknesses of SnO_2_ and PPy thin films.

**Figure 4 sensors-22-07252-f004:**
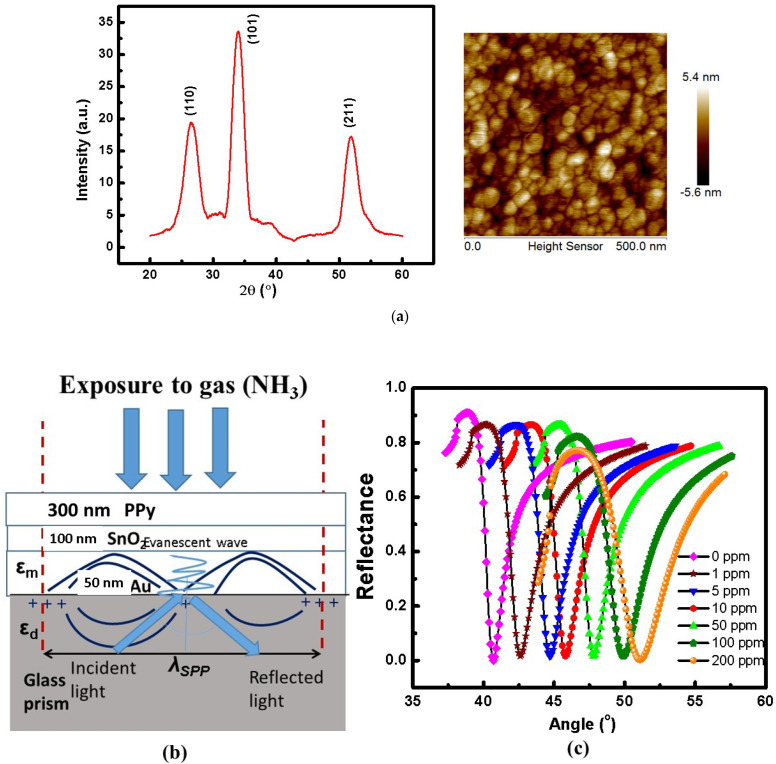
(**a**) Characterisation studies of SnO_2_/PPy nanocomposite thin film including XRD and AFM studies, (**b**) Schematic of the gas sensor system, (**c**) SPR reflectance measurements were made for the (prism + Au + SnO_2_ + PPy + NH_3_) Multi-layered structure after exposure to NH_3_ gas at various concentrations.

**Figure 5 sensors-22-07252-f005:**
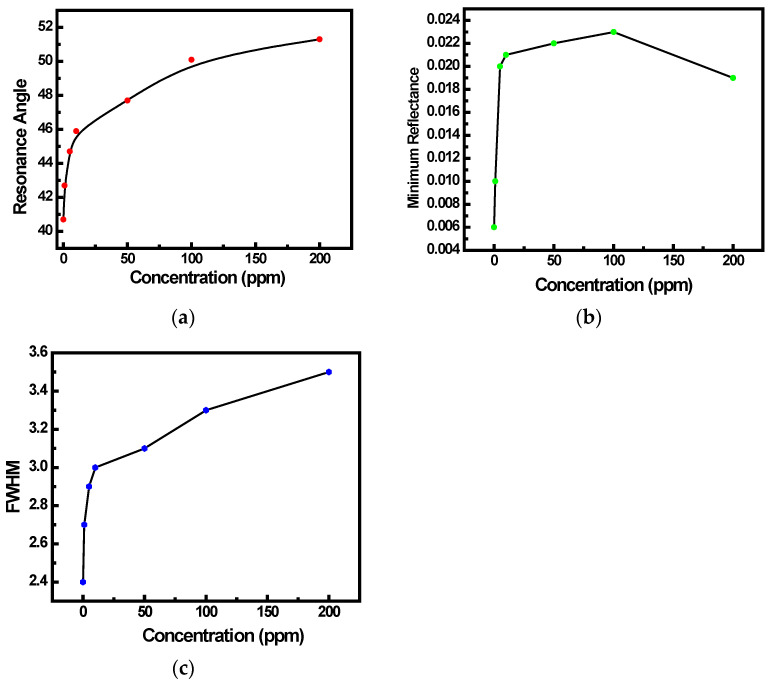
Variation of (**a**) SPR dip angle (θ_SPR_), (**b**) R_min_, and (**c**) FWHM for the SPR sensing system with different concentration of NH_3_ gas.

**Figure 6 sensors-22-07252-f006:**
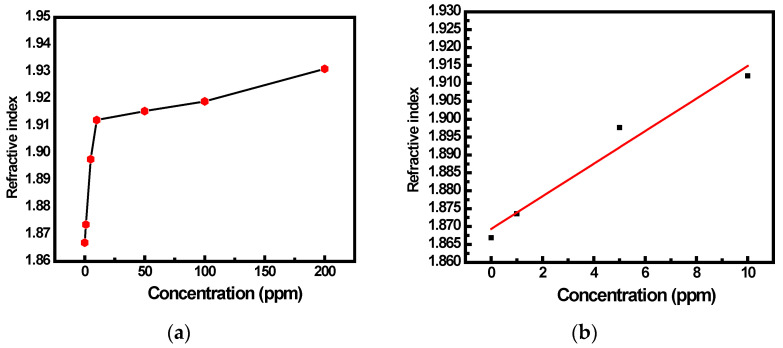
(**a**) Changes in the effective index of the (SnO_2_ + PPy) interface caused by the adsorption of NH_3_ gas with increasing concentration, and (**b**) changes in the refractive index up to a concentration of 10 ppm.

**Figure 7 sensors-22-07252-f007:**
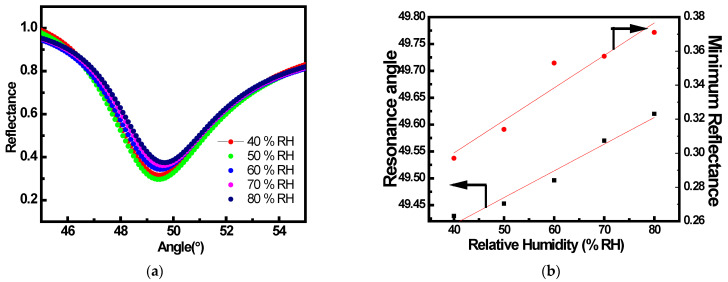
(**a**) SPR reflectance curves for prism/Au/SnO_2_/PPy structure on exposure to 10 ppm concentration of NH_3_ gas under varying humidity conditions, and (**b**) Resonance angle (θspr)(as shown by red dot) and Minimum Reflected Intensity (Rmin)(as shown by black box) variation with relative humidity.

**Figure 8 sensors-22-07252-f008:**
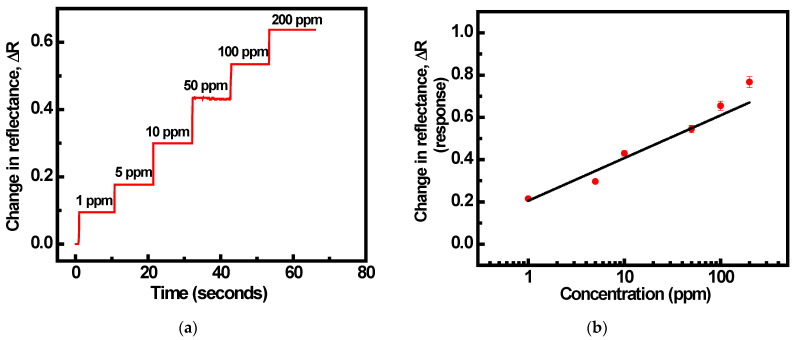
(**a**) Transient response of SPR sensor (prism + Au + SnO_2_ + PPy) system, (**b**) the calibration curve of the SPR sensor towards NH_3_ gas, and linear variation of change in reflectance with increasing concentration of NH_3_ gas.

**Figure 9 sensors-22-07252-f009:**
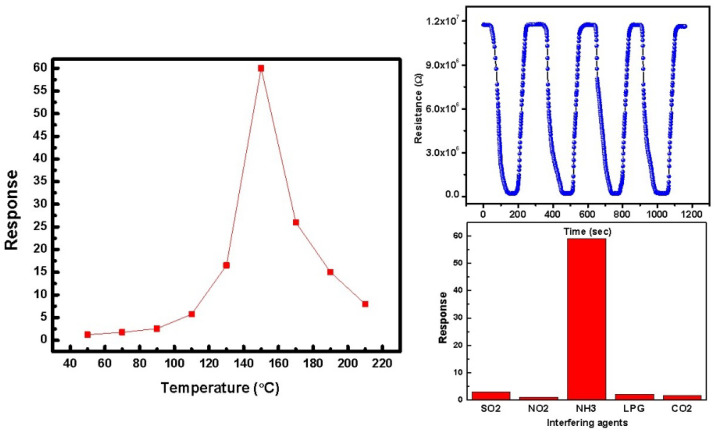
Response characteristics of the prepared sensor.

**Table 1 sensors-22-07252-t001:** Values of FWHM with thickness of SnO_2_ thin film.

Thickness of SnO_2_ (in nm)	Full Width Half Maximum (FWHM)
100	1.2
200 300 400	2.0 3.8 7.2

**Table 2 sensors-22-07252-t002:** Values of FWHM with thickness of PPy thin film.

Thickness of PPy (in nm)	Full Width Half Maximum (FWHM)
100	2.1
200 300 400	3.0 1.5 4.0

## Data Availability

Not applicable.

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
