# Peer review of "Exploitation of SnO2/Polypyrrole Interface for Detection of Ammonia Vapors Using Conductometric and Optical Techniques: A Theoretical and Experimental Analysis"

_sensors, 2022, doi:10.3390/s22197252_

Round 1

Reviewer 1 Report

The authors describe a novel thin film SnO2 sensor for ammonia detection. The advantages of the article:

1) The authors fabricated the thin film SnO2/Polypyrrole sensor.

2) The authors optimized the parameters of the sensor.

3) The authors measured and assessed the performance of the sensor.

Disadvantages of the article:

1) The reviewer has a problem identifying the theoretical part of the article. Maybe figure 3 was obtained by theoretical means. But 1a) The reader cannot easily identify the theoretical backgrounds of the graphs included in the figure. 1b) Small labels and legend in the figure does not help the reader. 1c) The figure caption on the next page is disappointing. The theoretical part of the article should be clearly written and vivid for the reader.

2) The reader is not sure about angle units used in figure 3 (right part) and 5.

3) The sensor performance is not compared to other ones known from the literature.

4) The left part of figure 8 lacks units on the Y-axis. The lower right part of the figure contains the X-axis label of the upper right part of the figure. This misleads the reader.

5) The reviewer cannot decide if the response of the sensor presented on the lower right part of figure 8 was obtained for the same concentration of the interfacing agents. It is not clear in what units the response was measured. Are they seconds as the label in the upper part of the graph suggests? 

Reviewer 2 Report

This work reports exploitation of sno2/polypyrrole interface for detection of ammonia vapours using conductometric and optical techniques. The SPR gas sensor is based on the change in refractive index at the Tin 12 Oxide (SnO2)/Polypyrrole (PPy) interface with the Ammonia (NH3) gas adsorption. The thickness of SnO2 and Polypyrrole thin films was optimized using theoretical calculations for a sharp SPR reflectance curve. For conductometric gas sensing studies, the film having optimized thicknesses for sharp SPR reflectance curves was separately prepared on IDTs. The work is interesting and can be of interest to sensor community, however some important discussion and investigations are missing, which are highlighted below.

The abstract is very generic and can be improved further. In the abstract the words “this publication and manuscript” is used, which can be replaced with “this work or contribution”

Inter-Digital Transducers should be corrected, to me interdigitated electrodes will be a right word.

The introduction section seems weak, beside inorganic and polymers, small organic materials has also been used for ammonia sensing. The author should compare the properties of inorganic with small organic materials and add some relevant papers (Sensors and Actuators B: Chemical 254 (2018): 805-810 and Journal of Materials Chemistry C 10 (4), 1326-1333, 2022) to broaden the introduction of the paper.

The surface morphology influence the sensing performance, the authors should investigate the morphology through AFM or SEM and correlate the morphology with the sensing performance. XRD analysis can be incorporated also.

In Fig. 8 it is shown the sensor showed high response to ammonia than other, can the authors comment on this, that why it is selective to ammonia than other gases?

The ambient humidity also affect the sensing performance, the authors should study the effect of sensing performance also.

Round 2

Reviewer 2 Report

The revised version reflects very well the feedback.